# Low-Carbohydrate Diet Inhibits Different Advanced Glycation End Products in Kidney Depending on Lipid Composition but Causes Adverse Morphological Changes in a Non-Obese Model Mice

**DOI:** 10.3390/nu11112801

**Published:** 2019-11-16

**Authors:** Tomoko Kaburagi, Kazuma Kanaki, Yuko Otsuka, Rumi Hino

**Affiliations:** 1Department of Health Science, Daito Bunka University, Saitama 355-8681, Japanotsuka@ic.daito.ac.jp (Y.O.); rumih0301@ic.daito.ac.jp (R.H.); 2Graduate School of Sports and Health Science, Daito Bunka University, Saitama 355-8681, Japan

**Keywords:** low-carbohydrate diet, non-obese mice, medium-chain triglyceride, kidney, advanced glycation end products

## Abstract

Low carbohydrate diets (LC diets) have been noted for adverse health effects. In addition, the effect of lipid composition on an LC diet is unclear. In this study, we used an LC diet containing two different lipids, lard (LC group) and medium-chain triglyceride oil (MCT-LC group), to examine the effect of an LC diet in non-obese mice. Male C57BL/6J mice were fed the control diet or one of the experimental diets ad libitum for 13 weeks. Increased renal weight and glomerular hypertrophy, as well as enlargement of intraglomerular small vessels with wall thickening, were seen in the LC and MCT-LC groups. Renal AMP-activated protein kinase activity was significantly decreased only in the LC diet group. On the other hand, epididymal adipose tissue weight and adipocyte area were markedly decreased only in the MCT-LC group. A positive effect was also observed in the kidney, where different advanced glycation end products, Nε-(carboxyethyl)-lysine and Nε-(carboxymethyl)-lysine, were inhibited depending on the lipid composition of the LC diet. Our findings suggest that, in non-obese conditions, low dietary intake of carbohydrates had both positive and negative impacts. The safety of diets low in carbohydrates, including the effects of fatty acid composition, requires further investigation.

## 1. Introduction

The effects and safety of a low-carbohydrate (LC) diet have been extensively studied [1,2,3,4,5]. Most previous studies have specifically focused on obese and diabetic conditions, demonstrating the effects of an LC diet on weight loss and blood glucose and lipid level regulation [6,7,8,9]. Conversely, other studies have investigated the negative health effects of an LC diet, such as increased oxidative stress, arteriosclerosis, and impaired insulin and glycemic responses [10,11,12,13]. LC diets may affect the development of various diseases, although the association of an LC diet with these various outcomes under obese conditions remains controversial. In obese conditions, adipocyte hypertrophy causes changes in adipocytokines [14], which in turn results in abnormalities in glucose and lipid metabolism [15]; however, abnormal metabolic states are less common in non-obese conditions, so it is doubtful that an LC diet in non-obese conditions will produce the same changes observed in studies targeting obese conditions. Although studies on the effects of an LC diet in non-obese states are limited, the morbidity of diabetes and heart disease, and mortality have been reported to increase with LC diet intake in several cohort studies [12,13,16]. A more detailed reported that an LC diet increased oxidative stress in non-obese mice by reducing hepatic superoxide dismutase activity, and also reduced lipid energy metabolism, but had no significant weight loss effect or effect on the regulation of blood glucose level [10,17].

The effects of different lipid compositions of an LC diet remain unclear, albeit that an LC diet is accompanied by an increase in lipid intake [1]. In a previous cohort study, a high level of meat lipid intake was reported to contribute to the development of heart disease in subjects on an LC diet [12,13,14]. In the present study, we examined the effect of an LC diet in non-obese mice using an LC diet that can be implemented in the long term, which contained 20% carbohydrates and added two types of lipid, lard, or medium-chain triglyceride (MCT) oil. Lard is a common meat lipid that is rich in long-chain saturated and monounsaturated fatty acids. These are major components of the western pattern diet and known to lead to obesity that compromises cardiovascular health [14,18]. MCT is a lipid composed of 8 to 10 carbon atoms, abundant in coconut oil and palm oil. MCT is absorbed directly from blood vessels and rapidly undergoes β-oxidation, thereby supporting the biosynthesis of ketone bodies. MCT oil, which induces a ketogenic condition, has been used to treat Alzheimer’s disease and epilepsy [19,20] and has also been reported to have antitumor effects [21]. In recent years, attention has also focused on diets rich in MCTs for reducing body fat in healthy people and for improving energy efficiency during sports [22,23,24]. However, ketogenic conditions have also been shown to have negative effects. It has been reported that acetoacetic acid, a kind of ketone body, promotes the production of reactive oxygen species [25] and involves protein glycation to produce N(epsilon)-(carboxyethyl)lysine (CEL), one of the advanced glycation end products (AGEs). Therefore, differences in lipid composition may have different effects in an LC diet. In addition, high fat and/or high protein intake from LC diets often cause chronic damage and inflammation of the glomeruli and tubule cells of the kidney, which is due to decreased kidney lipid metabolism and autophagy impairment. Proteinuria via impaired such renal function leads to the accumulation of AGEs [26,27,28,29,30,31].

In this study, we analyzed the effects of intake of an LC diet rich in lard or MCT oil on renal AGE production, morphology, and lipid metabolism of the kidneys in non-obese mice.

## 2. Materials and Methods

### 2.1. Materials

The following antibodies were used for histological immunostaining and western blot analysis: anti- Nε-(carboxymethyl)-lysine (#KH001) from TransGenic Inc. (Fukuoka, Japan), HRP-linked anti-rabbit IgG (#NA931) from General Electric Healthcare Biosciences (Piscataway, NJ, USA), anti-AMP-activated protein kinase (AMPK)α (#5832), anti-*p*-AMPK (#2535), and anti-β acting (#3662) from Cell Signaling Technology (Danvers, MA, USA).

### 2.2. Animals and Diet

Fifteen six-week-old male C57BL/6J mice were obtained from Charles River Laboratory, Inc. (Kanagawa, Japan) and randomly housed with five animals per cage in a temperature-controlled room at 24 °C with a 12 h light-dark cycle (lights off at 21:00). They were fed a control diet (64% carbohydrate, 16% fat, and 20% protein diet, based on AIN-93G, D10012G; Research Diets, Inc., New Brunswick, NJ, USA) during a one-week acclimation period, then divided into the three experimental diet groups of five mice each, namely an LC diet group, MCT-LC diet group, and control diet group. The groups were fed for 13 weeks. The composition of the experimental diet groups is shown in Appendix A. The carbohydrate content of the LC diet was 20% and was modified using casein as a protein source and high containing lard or MCT oil as a fat source. Mineral content was adjusted to the same level in all experimental diet groups. The mice were allowed free access to food and drinking water throughout the acclimation and experimental periods. Individual mouse weights and food consumption rates per cage were recorded twice weekly. This study was approved by the Daito Bunka University Animal Experiment Committee at Daito Bunka University in Saitama, Japan and complied with the guidelines of the Japanese Council on Animal Research.

### 2.3. Serum and Tissue Sampling

At the end of the 13-week feeding period, the mice fasted for 4 h then were anesthetized by isoflurane inhalation. Blood was sampled through the orbital cavity and separated into the serum by centrifugation. They were then sacrificed by cervical dislocation under anesthesia, and the livers, kidneys, and epididymal adipose tissues (EATs) were immediately removed and weighed. The livers and kidneys were homogenized with TBS containing 1% Triton X-100, a protease inhibitor cocktail (Roche Applied Science, Mannheim, Germany), and phosphatase inhibitors (Roche Applied Science, Mannheim, Germany). The samples were then centrifuged at 5000 rpm for 15 min, the supernatants were removed, and protein concentrations were measured using the BCA protein assay (Thermo Fisher Scientific, Waltham, MA, USA).

### 2.4. Western Blot Analysis

Proteins from the liver and kidney homogenate supernatants were separated with 5–20% gradient SDS-PAGE using the Laemmli buffer system and then transferred to nitrocellulose membranes. After blocking with 5% non-fat dry milk in TBS containing 0.05% Tween 20, the membranes were incubated with primary antibodies, diluted with the blocking buffer, and followed by HRP-conjugated secondary antibodies. Proteins were detected with Western lightning ECL Pro (PerkinElmer, Waltham, MA, USA) using an Image Quant LAS-4000mini (GE Healthcare Life Science, Issaquah, WA, USA).

### 2.5. Morphological Analysis

For morphological analysis, kidneys, livers, and EATs were fixed with paraformaldehyde, embedded in paraffin, sectioned at a thickness of 3 μm, and stained with hematoxylin-eosin (HE) and Periodic acid-Schiff (PAS). Sections were then examined microscopically. Morphological evaluations were conducted in a blinded manner using image analysis software Image J (NIH, Bethesda, MD, USA). Adipocytes were measured in 50 cell areas and glomeruli were measured for 20 areas per mouse randomly. The average value of each group was then statistically calculated. The kidney was also stained with a biotinylated anti-CML antibody. Briefly, kidney sections were incubated with 0.3% hydrogen peroxide in methanol and treated with the blocking solutions supplied in a tyramide signal amplification kit (TSA Biotin System; PerkinElmer Life Sciences), then incubated with biotinylated anti-CML antibody (overnight at 4 °C) and DAB-labeled streptavidin (30 min at room temperature).

### 2.6. Biochemical Measurements

Blood glucose and β-hydroxybutyric acid levels were measured in casual blood or after 4 h of fasting blood, obtained from the tail veins of anesthetized mice and analyzed using a FreeStyle Precision Neo (Abbott Laboratories, Chicago, IL, USA). Insulin, leptin, and adiponectin levels in fasting serum were measured using a mouse insulin ELISA kit, leptin ELISA kit (Shibayagi, Osaka, Japan) and adiponectin ELISA kit (Iwai Chemicals, Tokyo, Japan), respectively.

### 2.7. N(epsilon)-(carboxymethyl)lysine (CML) and N(epsilon)-(carboxyethyl) (CEL) Levels in Kidney and Serum

CML and CEL were measured using ELISA kits (OxiSelect™ CML ELISA Kit and OxiSelect™ CEL ELISA Kit, Cell Biolabs Inc., San Diego, CA, USA). Fasting serum and renal homogenate supernatant were diluted and tested in duplicate. The assays were performed according to the manufacturer’s instructions.

### 2.8. Statistical Analysis

Results are expressed as mean ± standard error of measurement (SEM). When one-way ANOVA was statistically significant, a multiple comparison test was used to make comparisons with each group using Tukey’s honestly significant difference (HSD) test. All *p* values <0.05 were considered statistically significant. All statistical calculations were performed using SPSS version 25.0 (IBM, Armonk, NY, USA).

## 3. Results

### 3.1. Body Weight and Weights of Liver and Kidney

Across 13 weeks with the experimental diet, body weight increased in all groups (from 23.0–27.9 g, 23.0–26.8 g, and 23.4–26.1 g in the LC, MCT-LC, and control groups, respectively), with no difference between groups at any time point (Figure 1). Experimental dietary intake also showed no difference between groups during the experiment. Weights of the liver and kidney were measured in the three diet groups and analyzed per gram of body weight (Figure 1). Kidney weight was greater in the LC and LC-MCT diet groups compared to the control diet group (12.2 and 12.0 vs. 10.5 mg per gram of body weight; *p* = 0.036 and *p* = 0.044, respectively).

### 3.2. Weights of EATs and Adipocyte Area.

EAT weight (per gram of body weight) was measured as an index of accumulation of visceral fat. Weight of EAT was lower in mice receiving the MCT-LC diet compared with the control diet (20.5 vs. 30.3 mg/g body weight; *p <* 0.05), whereas there was no difference between the LC and control diets (28.0 vs. 30.3; *p* = 0.776, Figure 2A). Morphological evaluation of EATs, stained by HE, showed significant inhibition of hypertrophy in the MCT-LC group but not among those assigned the LC diet (Figure 2B,C).

### 3.3. Morphological Analysis of Kidney

The renal morphological observation was performed on slices stained using HE and PAS. Glomerulus areas of LC and MCT-LC group were significantly larger than those of the control groups (Figure 3C). In addition, enlargement of intraglomerular small vessels, and imported arterioles hypertrophy with vessel wall thickness, a symptom of diabetic nephropathy and aging, was also observed in the LC and MCT-LC groups (Figure 3B, white arrows indicate enlargement of small vessels, black arrows indicate imported arterioles hypertrophy with vessel wall thickness ). Of the 20 glomeruli randomly counted per mouse, there were seven or more in the LC group and three or more in the MCT group that had enlargement of intraglomerular small vessels with wall thickening.

### 3.4. CEL and CML Levels in Serum and Kidney

We measured CEL and CML in kidney homogenate supernatant, as assessments of AGEs accumulation (Figure 4). Renal CML levels were markedly lower in the LC diet group than both the MCT-LC diet group and the control diet group (both *p <* 0.01, Figure 4A). Immunostaining of CML was found in the distal renal tubules, which was also less accumulated in the LC diet group than in both the MCT-LC diet group and the control diet group (Figure 4B). Conversely, CEL was significantly lower in the MCT-LC diet group than the LC diet group and the control group (both *p <* 0.05, Figure 4A). Serum CML and CEL could not be detected, because concentrations were very low.

### 3.5. Biochemical Analysis

Casual blood β-hydroxybutyrate was higher in the MCT-LC diet (0.45 ± 0.12) than the control diet group (0.08 ± 0.01, *p <* 0.05) and the LC (0.31 ± 0.03, *p* = 0.18). Fasting blood β-hydroxybutyrate also tended to be higher in the MCT-LC diet group (1.72 ± 0.18) than the LC (1.65 ± 0.17) and control (1.48 ± 0.08) diet groups. Both casual and fasting blood glucose, though not statistically significant, showed a tendency to be higher in the LC diet group than control and MCT-LC diet groups (*p* = 0.09 and *p* = 0.06, respectively of casual serum, Table 1). Fasting serum adiponectin, a marker of insulin sensitivity, was significantly lower in the LC (*p <* 0.05) and MCT-LC diet (*p <* 0.01) groups than the control diet group. Fasting serum leptin, another marker of insulin resistance, was lower in the MCT-LC diet groups than the control group, but the differences were not significant (Figure 5). Fasting serum insulin was lower in the MCT-LC group than the control diet group (*p <* 0.05).

### 3.6. Hepatic Glycogen Accumulation

The PAS-positive part was evaluated as glycogen accumulation. Glycogen was observed to have accumulated overall in the control group (Figure 6). Particularly little accumulation was observed in the LC group. Partial accumulation of glycogen was observed in the MCT-LC group.

### 3.7. AMP-Activated Protein Kinase (AMPK) Activation in Kidney and Liver

Renal phosphorylated AMPK were significantly reduced only in the LC diet group in comparison to the control group (*p <* 0.05, Figure 7A,C). In AMPK activation of the hepatic sample, we observed a tendency toward reduction in the LC diet group, as well as recovery of phosphorylation in the MCT-LC group (Figure 7B,D). Downregulation of *p*-AMPK indicated that fatty acid metabolism was suppressed [17].

## 4. Discussion

In this study, we analyzed the impact of a diet low in carbohydrate and high in one of two different lipids, lard (LC diet) or MCT oil (MCT-LC diet), on renal morphology and AGEs production, and AMPK activity in kidney and liver using an experimental mouse non-obese model. Six-week-old male C57BL/6J mice were fed a control diet or one of the experimental diets ad libitum for 13 weeks. While there were no significant differences in body weight among the diet groups, weights of EATs and adipocyte area were significantly lower only in mice fed the MCT-LC diet compared to the control diet. Further, we observed enlargement of small vessels with wall thickening in a part of the intraglomeruli and imported arterioles hypertrophy, with a greater increase in renal weight in the LC and MCT-LC groups, findings which are suggestive of harmful effects of LC and MCT-LC diets. Serum adiponectin was significantly low in the LC and MCT-LC diet groups, and blood glucose tended to be higher only in the LC diet group (*p* = 0.09 vs. control, *p* = 0.06 vs. MCT-LC). In addition, hepatic glycogen accumulation and AMPK activation were markedly decreased in the LC group.]. On the other hand, the production of different AGEs was decreased due to the lipid composition of diets low in carbohydrates. These results suggest that a diet low in carbohydrate has both a negative and positive effect on the kidney function depending on lipid composition, which might be related to glucose and lipid metabolism.

The present study is the first to show that the intake of a diet in low carbohydrate inhibited renal AGEs production while adversely affecting renal morphological changes. The kidney plays an important role in controlling the excretion and accumulation of AGEs. Acetone, metabolites formed from ketone bodies, known to produce CEL, which induce diabetic complications in the kidney and lens [26]. The MCT-LC diet, which confers a high ketone state, was speculated to have higher CEL accumulation. On the contrary, however, our results showed that CEL was significantly lower in the MCT-LC group, suggesting that an MCT-LC diet may have a positive effect on renal function. The possibility of a positive effect of inhibiting renal damage in the MCT-LC diet group is an important finding, given that nephropathy is a common sequela of diabetes. In the MCT-LC group, casual serum showed high levels of β-hydroxybutyrate but not fasting serum. Furthermore, in the liver of the MCT-LC group, glucose accumulation was shown slightly higher than that in the LC diet, and AMPK activity was also maintained. These results may arise from the rapid metabolism of fasting β-hydroxybutyrate, with the MCT-LC diet promoting lipid metabolism through β-oxidation. On the other hand, CML was markedly lower in the LC diet group compared to the control group (*p <* 0.05), in which nephrotic distal tubules showed abundant CML, especially in the control diet group. CML had been known to accumulate in the proximal tubule by binding with megalin, the proximal tubule’s membrane receptor [32,33,34]. The accumulation of CML in the distal renal tubules is a new finding, as seen in the control group. The Klotho gene known as involved in the suppression of renal damage and strongly expressed in the distal renal tubule may be involved [35]. CML, which is generated via a different route to CEL, is a common AGE which is thought to be elevated by high blood sugar levels. This is inconsistent with our findings; however, while the between-group differences in blood glucose were not significant, point estimates for the LC diet group (casual serum glucose: 333.8 ± 23.8; fasting serum glucose: 203.2 ± 21.9) were higher than those for the MCT-LC diet (casual serum glucose: 264.9 ± 37.7; fasting serum glucose: 141.2 ± 7.1) or control diet (casual serum glucose: 300.5 ± 9.2; fasting serum glucose: 185.5 ± 25.5) groups. Blood glucose has also been noted to be elevated in other reports of long-term low-carbohydrate diets [36]. Handa et al. reported that gluconeogenesis-related enzymes (e.g., Phosphoenolpyruvate carboxykinase and G6Pase) were decreased, and hepatic STAT3, which plays a role in the inhibition of gluconeogenesis-related enzymes increase, might account for the higher glucose levels observed in LC diets [36]. Further study is needed to assess the CML inhibition mechanisms of LC diets, including the production of CML precursors, such as methylglyoxal and lipid peroxide.

Adiponectin and leptin, produced by adipocytes, are known as adipokines. Adiponectin plays an inhibition role in the development of insulin resistance [37]. Leptin is a hormone involved in regulating appetite, body weight, glycemia, and neuroendocrine function [38]. We found that both the LC and MCT-LC diets led to significant decreases in adiponectin compared to the control diet. Serum leptin concentrations were lower in the MCT-LC diet groups than the control diet group. The trend of high blood glucose and low insulin was observed in the LC group. Although these data showed no statistical difference, that might be speculated the impairment of insulin secretion due to β-cell damage of the pancreas due to the high fat intake of the LC diet [39]. Given that adiponectin and leptin increase as fat cells get smaller [36], our finding of reduced adipocytokines (adiponectin and leptin) despite the reduction in adipose tissue and adipocyte area under the MCT-LC diet condition warrants further investigation.

Morphology of renal tissue showed a significantly larger glomerular area, enlargement of intraglomerular small vessels with wall thickening and imported arterioles hypertrophy, which are symptoms of diabetic nephropathy and aging [31,32], in the LC and MCT-LC diet groups than the control diet. Furthermore, renal AMPK phosphorylation was significantly reduced in the LC group, suggesting the possibility of renal dysfunction due to decreased AMPK activity [26]. To clarify renal dysfunction, serum and urine biochemical markers, such as creatinine and urea, and histological fibrosis markers need to be further examined. The glomerular morphological disorder is an established microvascular complication of insulin dysregulation and diabetes and is thought to be related to the proliferation of AGEs [33,34], though CML and CEL production was not associated with morphological changes in our study. In addition, AGEs, the production of which is known to be stimulated by hyperglycemia, showed a significant decrease in CML in both renal immunohistology and biochemical analysis in the LC group, whereas blood glucose levels tended to be higher, which requires further analysis. With regard to the morphological investigation, the adipocyte area was significantly smaller in the MCT-LC diet group than in the control diet group. The LC diet group had a slightly smaller adipocyte area than the control group, although the difference was not significant. Adipocyte hypertrophy is thought to play a central role in the pathogenesis of many obesity-related disorders, including insulin resistance and diabetes [15,37,40]. Likewise, adipocyte hypertrophy and adipocyte dysregulation are suggested to reduce insulin sensitivity even in non-obese individuals [41]. In the liver, the LC group tended to be low AMPK activation and glucose accumulation compared to the MCT-LC group. This indicates that lipid energy metabolism was increased only in the MCT-LC group during low-carbohydrate status. Thus, in the present non-obese conditions, we showed that an inhibitory effect on adipocytes hypertrophy with obesity and obesity-related disorders occurred only in the LC diet high in MCT. To clarify the effects on glucose and lipid metabolism in the lipid composition of the LC diet, further studies are needed.

## 5. Conclusions

This study shows that, in non-obese conditions, low dietary intake of carbohydrates had both positive and negative impacts. Further, the positive impact differed by lipid composition: an LC diet with high MCT showed anti-obesity effects, including a reduction of adipose tissue and adipose area, and renal CEL reduction, whereas an LC diet high in lard resulted in a large reduction in renal CML production. Regarding negative effects, in contrast, this study shows that low dietary intake of carbohydrate may have harmful sequelae, such as glomerular morphological disorder and a decrease in adiponectin. In our study present study, LC and MCT-LC diets were not completely safe, having both good and bad aspects. We also found that renal AGE production was strongly influenced by the lipid composition of the LC diet, which is an important discovery for disease prevention. The safety of diets low in carbohydrates, including the effects of fatty acid composition, requires further investigation.

## Figures and Tables

**Figure 1 nutrients-11-02801-f001:**
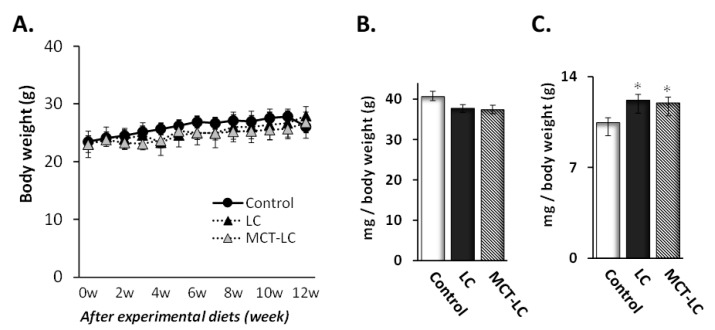
Body weights (**A**), liver weights (per gram of bodyweight) (**B**), and kidney weights (per gram of bodyweight) (**C**) of mice receiving experimental and control diets. Data shown are mean ± SEM. Control, control diet; LC, low-carbohydrate diet high in lard; MCT-LC, low-carbohydrate diet high in medium-chain triglyceride. * *p <* 0.05 vs. control by the Tukey’s HSD test.

**Figure 2 nutrients-11-02801-f002:**
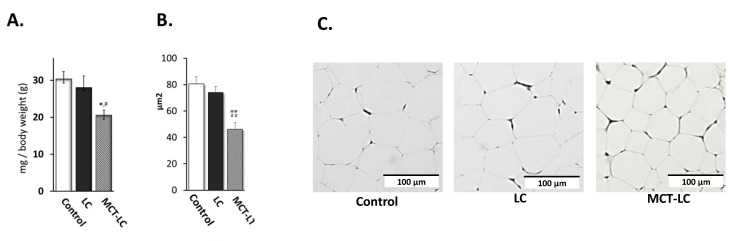
Weights (per gram of body weight) (**A**), adipocyte area (**B**), and hematoxylin-eosin (HE) staining (**C**) of epididymal adipose tissues in the three diet groups. Data shown are mean ± SEM. Control, control diet; LC, low-carbohydrate diet high in lard; MCT-LC, low-carbohydrate diet high in medium-chain triglyceride. * *p <* 0.05, ** *p <* 0.01 vs. control, # *p <* 0.05, ## *p <* 0.01 vs. LC by the Tukey’s HSD test.

**Figure 3 nutrients-11-02801-f003:**
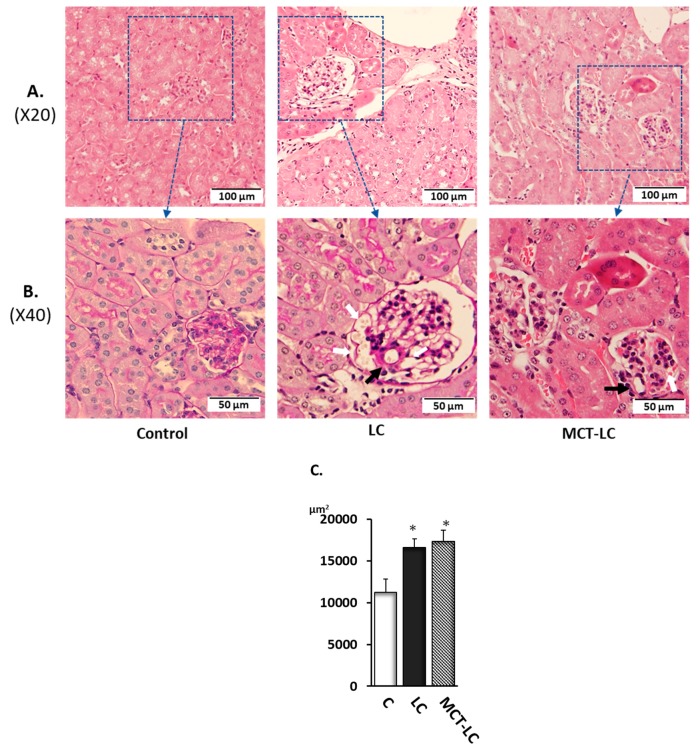
Renal morphological observation by HE staining (**A**), Periodic acid-Schiff staining (**B**) (white arrows indicate enlargement of small vessels, black arrows indicate imported arterioles hypertrophy with vessel wall thickness ), and measurement of glomerular area (**C**). Data shown are mean ± SEM. Control, control diet; LC, low-carbohydrate diet high in lard; MCT-LC, low-carbohydrate diet high in medium-chain triglyceride. * *p <* 0.05 vs. control by Tukey’s HSD test.

**Figure 4 nutrients-11-02801-f004:**
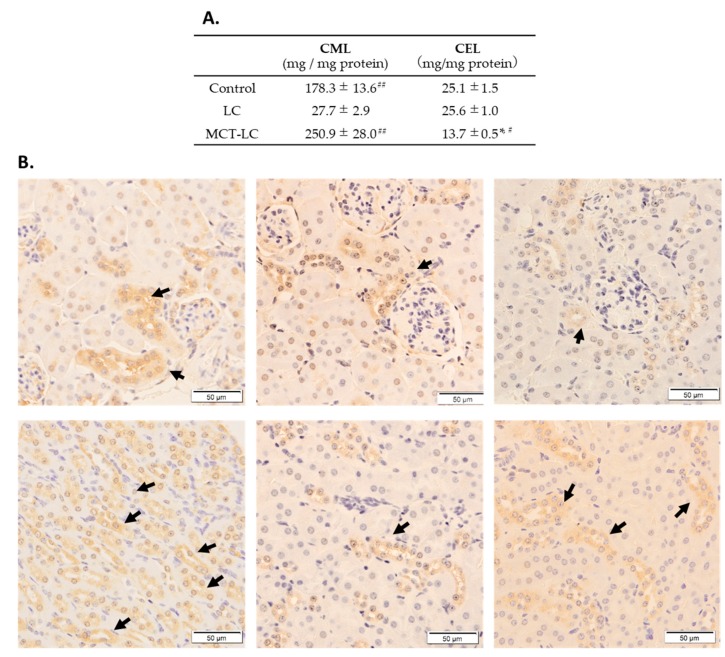
CML and CEL concentrations in kidney supernatant (**A**) and CML staining of the kidney (**B**), The first row (**B**) shows the renal cortex area where glomeruli gather, and the second row (**B**) shows medulla area where tubules gather. Data shown are mean ± SEM. Control, control diet; LC, low-carbohydrate diet high in lard; MCT-LC, low-carbohydrate diet high in medium-chain triglyceride. * *p* < 0.05, ** *p* < 0.01 vs. control, # *p* < 0.05, ## *p* < 0.01 vs. LC by the Tukey’s HSD test.

**Figure 5 nutrients-11-02801-f005:**
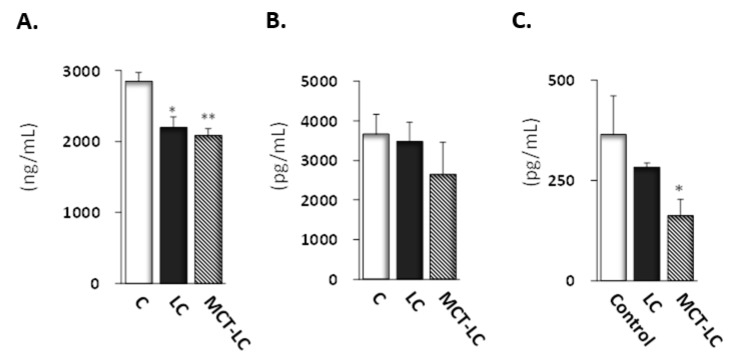
Adiponectin (**A**), leptin (**B**), and insulin (**C**) levels in fasting serum. Data shown are mean ± SEM. Control, control diet; LC, low-carbohydrate diet high in lard; MCT-LC, low-carbohydrate diet high in medium-chain triglyceride. * *p <* 0.05, ** *p <* 0.01 vs. control by the Tukey’s HSD test.

**Figure 6 nutrients-11-02801-f006:**
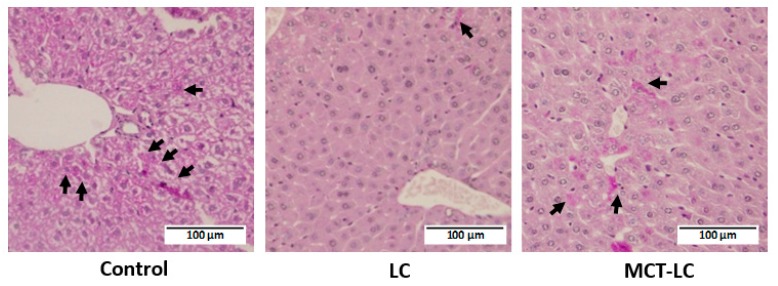
Hepatic glycogen accumulation by PAS staining. Arrows show glycogen accumulation area. Control, control diet; LC, low-carbohydrate diet high in lard; MCT-LC, low-carbohydrate diet high in medium-chain triglyceride.

**Figure 7 nutrients-11-02801-f007:**
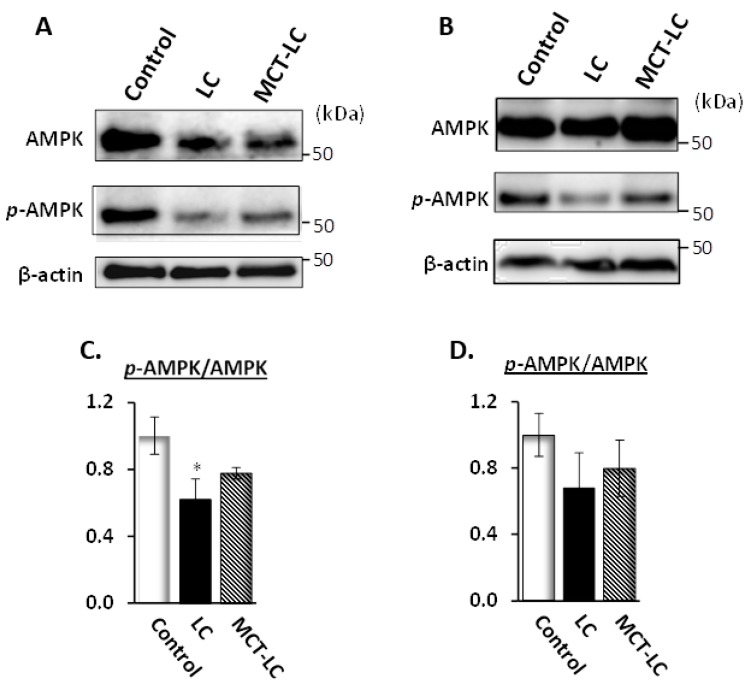
AMPK activation in renal sample (**A**,**C**) and hepatic sample (**B**,**D**). Data shown are mean ± SEM. Control, control diet; LC, low-carbohydrate diet high in lard; MCT-LC, low-carbohydrate diet high in medium-chain triglyceride. * *p <* 0.05 vs. control by Tukey’s HSD test.

**Table 1 nutrients-11-02801-t001:** Casual serum and fasting serum glucose and β-hydroxybutyrate.

	Casual Serum		Fasting Serum	
	Glucose (mg/dL)	β-Hydroxybutyrate (mmol/L)	Glucose (mg/dL)	β-Hydroxybutyrate (mmol/L)
Control	300.5 ± 9.15	0.08 ± 0.01	185.5 ± 25.50	1.48 ± 0.08
LC	333.8 ± 23.80	0.31 ± 0.03	204.2 ± 21.85	1.65 ± 0.17
MCT-LC	264.9 ± 37.74	0.45± 0.12 *	141.2 ± 7.09	1.72 ± 0.18

Values are mean ± SEM, *n* = 5. Control, control diet; LC, low-carbohydrate diet high in lard; MCT-LC, low-carbohydrate diet high in medium-chain triglyceride. * *p <* 0.05 vs. control by the Tukey’s HSD test.

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
