# Peer review of "Low-Carbohydrate Diet Inhibits Different Advanced Glycation End Products in Kidney Depending on Lipid Composition but Causes Adverse Morphological Changes in a Non-Obese Model Mice"

_nutrients, 2019, doi:10.3390/nu11112801_

Round 1

Reviewer 1 Report

Abstract: major changes required.

11 recommend rewording  “the effect of differences in lipid composition of LC diets is unclear”

18  Incorrect statement According to your serum adiponectin results there was no significant difference between the LC vs MCT group. Also, glucose was not significantly different for any of the groups

20 “renal dysfunction” This conclusion is not supported by results discussed in the abstract. Unsure how they reach it.

Results: Minor changes required

146 Kidney weight was proportionally greater since it is based on mg/g body weight not based on total weight. This should be also noted with the other organ weights.

147 No standard error of measurement reported

Figure 1 (C) the textbox with the asterisk is blocking the bars.

148 Should report only EAT. Visceral fat was not measured since the other visceral fat components were not collected.

165 states: CEL was significantly lower in the MCT-LC diet group than the LC diet group (p<0.05), while there was no difference between the MCT-LC diet group and the control group or between the LC diet group and the MCT-LC diet group. This can’t be correct

183 Can’t state that blood glucose was higher in LC when none of the groups was statistically different.

190.- Table . Those fasting glucose levels seem abnormally high for C57BL/6J mice. This test may not be reliable.

198-203 bold font

Discussion: Minor changes

215 Incorrect statement: Same as correction on the abstract. Serum adiponectin was significantly lower than control but higher than MCT-LC diet.

228 Incorrect statement LC diet did not result in increased AGE production

229 Why will lower CEL accumulation on the MCT-LC group will be associated with negative effects on renal function? Would not that be the opposite?

Conclusions: Major changes

273 Conclusions should be separate from LC and LC-MCT since they showed different results. In fact CML kidney staining showed less of this AGE on the LC diet when compared to the control.

Reviewer 2 Report

The study by Kaburagi et al. examines the effect of low-carbohydrate diet and different lipid compositions on body weight, glucose and adipokine production. It also attempted to make a link of these changes with renal AGEs deposition and histological abnormalities. The topic is new and interesting, but the data are fragmented and insufficient to support the conclusion.

Why is it important to study the effect of LC in non-obese population? What is the odd for someone without obesity but having a LC diet? These points need to be elaborated or otherwise the significance and clinical implication of the study will be questioned.

There is no background information about the effect of LC diet on kidney, which is a major point of this manuscript. It is also unclear why the author chose kidney to study? Throughout the manuscript, the link between LC diet, lipid metabolism and renal pathology is very unclear. There is no clear evidence or result to support the proposed mechanism via ketone bodies.

Sample size is quite small: only 5 per group. It is unclear why there are only 3 or 2 samples in western blot.

There is no explanation to why β-hydroxybutyric acid level was measured. What is the difference between casual and fasting serum levels and which one is more important?

There is no description of the measurement method for intraglomerular area. What renal pathology does this imply?

The authors discuss a lot about insulin sensitivity, but glucose intolerance and insulin levels were not measured. There are also no data of blood fatty acid and triglyceride levels.

There are not enough samples for WB, so any conclusion from this data is invalid.

Glomerular hyalinization? Where does this conclusion come from? Fibrosis markers such as collagen I and III as well as fibronectin need to be measured by IHC. Alternatively, PAS, Masson-Trichome, or PSR staining should be performed to assess the level of fibrosis. If the damage is mild, oxidative stress and inflammatory markers are needed.

None of the common markers for kidney damage, e.g. ACR, serum creatinine, BUN and KIM-1, were measured.

AGE level in kidney is not equal to renal AGE production but could be largely due to accumulation from blood stream.

The discussion about CEL and CML is confusing. It is unclear whether the author means an increase or decrease of each metabolite is good or bad.

Many results in this study are not significant, thus not supporting the conclusion that the authors are trying to make.

Minor issues:

Figure 1C: format issue

Figure 4B: What is the difference between the 1strow and 2ndrow of pictures?

Reviewer 3 Report

The title needs to be reformulated. The acronym AGE should be avoided since it is not a very common acronym. Also the statement that it is a non-obese mouse model is somewhat confusing since the authors in figure 2A, shows reduced fat weight in MCT-LC mice. One should expect the control mice to have normal body composition, so maybe the MCT-LC showed signs on starvation in adipose tissue. Having the whole body composition (lean mass vs fat mass) would be optional. In general, to introduce obesity in C57/Bl6 mice, high-fat diet feeding should not be started until the mice are well adult, since they seem to develop resistance to a higher fat content in the diets. Or the feeding intervention should >20 weeks. See https://www.physiology.org/doi/pdf/10.1152/ajpheart.00382.2017  The casual blood b-hydroxybutyrate was higher in the MCT-LC fed mice. What should be the cut off for having the mice in ketosis? A study by Nilsson et al., https://www.ncbi.nlm.nih.gov/pubmed/27891164 demonstrated higher plasma b-hydroxybutyrate in control chow fed female C57/Bl6J mice. Both the LC-diets should display ketosis or at least similar B-hydroxibutyrate values as a response to he low-carbohydrate diet. If the intention of the study was to see the effects of the MCT, a second control group given the same amount MCT in the control diet should have been included. In the introduction (lines 36-39), the authors discuss that changes in adipocytokines result in abnormal lipid and glucose metabolism. However, only blood glucose and b-hydroxybutyrate is measured. Blood glucose levels do not differ, so it is hard to say that this model has abnormal glucose and/or lipid metabolism. The LC and MCT-LC diets contained a larger proportion of proteins. Maybe glucose levels were not altered because of increased gluconeogenesis? Reduced renal function can induce insulin resistance. Insulin should have been measured. Glucose and protein in urine should have been measured as a marker of malfunctioning kidneys. Also, urea levels are discussed in the discussion part, line218, as a marker of renal dysfunction. This measurement should have been included as a marker of reduced kidney function. The mice were fed three different diets with different composition of the major energy sources. Did the authors control food intake? It would make huge sense to see the daily caloric intake throughout the study period. Casual serum is not described (ad libitum?), and what is the purpose of measuring both fasted and casual serum. The sections in fig 3 are blurry and the scale bar is unreadable. The method of measuring intraglomerular area is missing. Since AGEs can accumulate with megalin, co-staining CEL/CML with megalin would have improved the study. Regarding the western blot data, statistical power should be calculated before translating the data. In the discussion, line 213, hyalinization in the kidney is mentioned. How was it measured? The figure lay-outs needs to be improved. i.e fig 1C, the asterisk is partly covering the bars. Figure legends should be below the figures. This is not done in fig 2 or 3. Also the figure legends could provide more information about the data. Table 1 contains both bold and underlined text where it should not be such. Line 186; check the direction of >/<. Lines 164-168; The sentence is long and unclear.

Reviewer 4 Report

Dr. Kaburagi and her colleagues had demonstrated the possible harmful effect of a high-fat diet that combined with a low carbohydrate diet in a non-obese and non-DM mice model through the AGEs. The low carbohydrate diet is more and more popular at this time, especially for weight reduction. There were several studies that also provided the kidney damage via the different mechanisms.1,2,3 The following are some suggestions:

Did the authors find any dose-effect? For example, Using different LC/Carbohydrate percentage which might result in different level of CEL and CML. IF the mice with LC or MIC-LC eat the control diet after 13 weeks feeding, will the result attenuate?

Yamamoto T et al. J Am Soc Nephrol. 2017 May;28(5):1534-1551. Kuwahara S, et al. J Am Soc Nephrol. 2016 Jul;27(7):1996-2008. Declèves AE, et al. Kidney Int. 2014 Mar;85(3):611-23

Round 2

Reviewer 2 Report

Although I appreciate the authors' response to my review, many of the raised concerns were not fully addressed and I believe the quality of the data is still not up to the journal publication standard.